RESEARCH

# Sci-Hub provides access to nearly all scholarly literature

**Abstract** The website Sci-Hub enables users to download PDF versions of scholarly articles, including many articles that are paywalled at their journal's site. Sci-Hub has grown rapidly since its creation in 2011, but the extent of its coverage has been unclear. Here we report that, as of March 2017, Sci-Hub's database contains 68.9% of the 81.6 million scholarly articles registered with Crossref and 85.1% of articles published in toll access journals. We find that coverage varies by discipline and publisher, and that Sci-Hub preferentially covers popular, paywalled content. For toll access articles, we find that Sci-Hub provides greater coverage than the University of Pennsylvania, a major research university in the United States. Green open access to toll access articles via licit services, on the other hand, remains quite limited. Our interactive browser at https://greenelab.github.io/scihub allows users to explore these findings in more detail. For the first time, nearly all scholarly literature is available gratis to anyone with an Internet connection, suggesting the toll access business model may become unsustainable.

DOI: https://doi.org/10.7554/eLife.32822.001

DANIEL S HIMMELSTEIN*, ARIEL RODRIGUEZ ROMERO, JACOB G LEVERNIER, THOMAS ANTHONY MUNRO, STEPHEN REID MCLAUGHLIN, BASTIAN GRESHAKE TZOVARAS AND CASEY S GREENE*

## Introduction

Recent estimates suggest paywalls on the web limit access to three-quarters of scholarly literature (*Piwowar et al., 2018*; *Khabsa et al., 2014*; *Bosman and Kramer, 2018*). The open access movement strives to remedy this situation (*Tennant et al., 2016*). After decades of effort by the open access community (*Royster, 2016*), nearly 50% of newly published articles are available without paywalls (*Piwowar et al., 2018*; *Archambault et al., 2014*; *Van Noorden, 2013a*).

Despite these gains, access to scholarly literature remains a pressing global issue. Foremost, widespread subscription access remains restricted to institutions, such as universities or medical centers. Smaller institutions or those in the developing world often have poor access to scholarly literature (*Meadows, 2015*; *Kirsop and Chan, 2005*; *Bendezú-Quispe et al., 2016*). As a result, only a tiny percentage of the world's population has been able to access much of the scholarly literature, despite the fact that the underlying research was often publicly

or philanthropically funded. Compounding the problem is that publications have historically been the primary, if not sole, output of scholarship. Although copyright does not apply to ideas, journals leverage the copyright covering an article's prose, figures, and typesetting to effectively paywall its knowledge.

Since each article is unique, libraries cannot substitute one journal subscription for another without depriving their users of potentially crucial access. As a result, the price of journal subscriptions has grown at a faster rate than inflation for several decades (*Association of Research Libraries, 2017*), leading to an ever-present "serials crisis" that has pushed library budgets to their brink while diverting funds from other services (*Roth, 1990*). Meanwhile, publishing has trended towards oligopoly (*Larivière et al., 2015*), with nondisclosure clauses obfuscating price information among subscribers (*Bergstrom et al., 2014*) while publishers profit immensely (*Morrison, 2012*; *Buranyi, 2017*; *Van Noorden, 2013b*). Price increases have persisted over the last decade

**\*For correspondence:** daniel. himmelstein@gmail.com (DSH); greenescientist@gmail.com (CSG)

**Competing interests:** The authors declare that no competing interests exist.

(*Bosch and Henderson, 2017*; *Lawson et al., 2015*; *Lawson, 2017a*). For example, EBSCO estimates that per-journal subscription costs increased by 25% from 2013–2017, with annual subscription to a journal for research libraries now averaging $1,396 (*EBSCO, 2017*).

In this study, we use the term "toll access" (also known as "closed access") to refer to pay-walled literature (*Suber, 2017*). On the other hand, we refer to literature that is free to read as "open access". Furthermore, we discuss two variants of open access: "libre" and "gratis" (*Suber, 2017*; *Suber, 2008*). Libre open access refers to literature that is openly licensed to allow reuse. Gratis open access refers to litera-ture that is accessible free of charge, although permission barriers may remain (usually due to copyright) (*Himmelstein, 2016*).

The website Sci-Hub, now in its sixth year of existence, provides gratis access to scholarly lit-erature, despite the continued presence of pay-walls. Sci-Hub brands itself as "the first pirate website in the world to provide mass and public access to tens of millions of research papers." The website, started in 2011, is run by Alexandra Elbakyan, a graduate student and native of Kazakhstan who now resides in Russia (*Bohannon, 2016a*; *Schiermeier, 2015*). Elbakyan describes herself as motivated to pro-vide universal access to knowledge (*Elbakyan, 2016a*; *Elbakyan, 2015*; *Milova, 2017*).

Sci-Hub does not restrict itself to only openly licensed content. Instead, it retrieves and distrib-utes scholarly literature without regard to copy-right. Readers should note that, in many jurisdictions, use of Sci-Hub may constitute copyright infringement. Users of Sci-Hub do so at their own risk. This study is not an endorse-ment of using Sci-Hub, and its authors and pub-lishers accept no responsibility on behalf of readers. There is a possibility that Sci-Hub users — especially those not using privacy-enhancing services such as Tor — could have their usage history unmasked and face legal or reputational consequences.

Sci-Hub is currently served at domains includ-ing https://sci-hub.hk, https://sci-hub.la, https://sci-hub.mn, https://sci-hub.name, https://sci-hub.tv, and https://sci-hub.tw, as well as at sci-hub22266oqcxt.onion — a Tor Hidden Service (*Dingledine et al., 2004*). Elbakyan described the project's technical scope in July 2017 (*Elbakyan, 2017*): "Sci-Hub technically is by itself a repository, or a library if you like, and not a search engine for some other repository. But

of course, the most important part in Sci-Hub is not a repository, but the script that can down-load papers closed behind paywalls."

One method Sci-Hub uses to bypass paywalls is by obtaining leaked authentication credentials for educational institutions (*Elbakyan, 2017*). These credentials enable Sci-Hub to use institu-tional networks as proxies and gain subscription journal access. While the open access movement has progressed slowly (*Björk, 2017*), Sci-Hub represents a seismic shift in access to scholarly literature. Since its inception, Sci-Hub has expe-rienced sustained growth, with spikes in interest and awareness driven by legal proceedings, ser-vice outages, news coverage, and social media (*Figure 1* and *Figure 1—figure supplement 1*). Here we investigate the extent to which Sci-Hub provides access to scholarly literature. If Sci-Hub's coverage is sufficiently broad, then a radi-cal shift may be underway in how individuals access scholarly literature.

In *Figure 1*, The letters **A**, **B**, **C**… refer to the following events:

○ **A** Created by Alexandra Elbakyan, the Sci-Hub website goes live on September 5, 2011.

○ **B** Several LibGen domains go down when their registration expires, allegedly due to a longtime site administrator passing away from cancer.

○ **C** Elsevier files a civil suit against Sci-Hub and LibGen — at the respective domains sci-hub.org and libgen.org — in the U.S. District Court for the Southern District of New York (*Van der Sar, 2015a*; *DeMarco et al., 2015a*). The complaint seeks a "prayer for relief" that includes domain name seizure, damages, and "an order disgorging Defendants' profits".

○ **D** Elsevier is granted a preliminary injunc-tion to suspend domain names and restrain the site operators from distribut-ing Elsevier's copyrighted works (*Van der Sar, 2015b*; *Sweet, 2015*). Shortly after, Sci-Hub and LibGen resurface at alterna-tive domains outside of U.S. court jurisdic-tion, including on the dark web (*Schiermeier, 2015*; *Van der Sar, 2015c*).

○ **E** The article "Meet the Robin Hood of Sci-ence" by Simon Oxenham spurs a wave of attention and news coverage on Sci-Hub and Alexandra Elbakyan (*Oxenham, 2016*), culminating in *The New York Times* asking "Should all research papers be free?" (*Murphy, 2016*).

○ **F** The article "Who's downloading pirated papers? Everyone" by John Bohannon shows Sci-Hub is used worldwide,

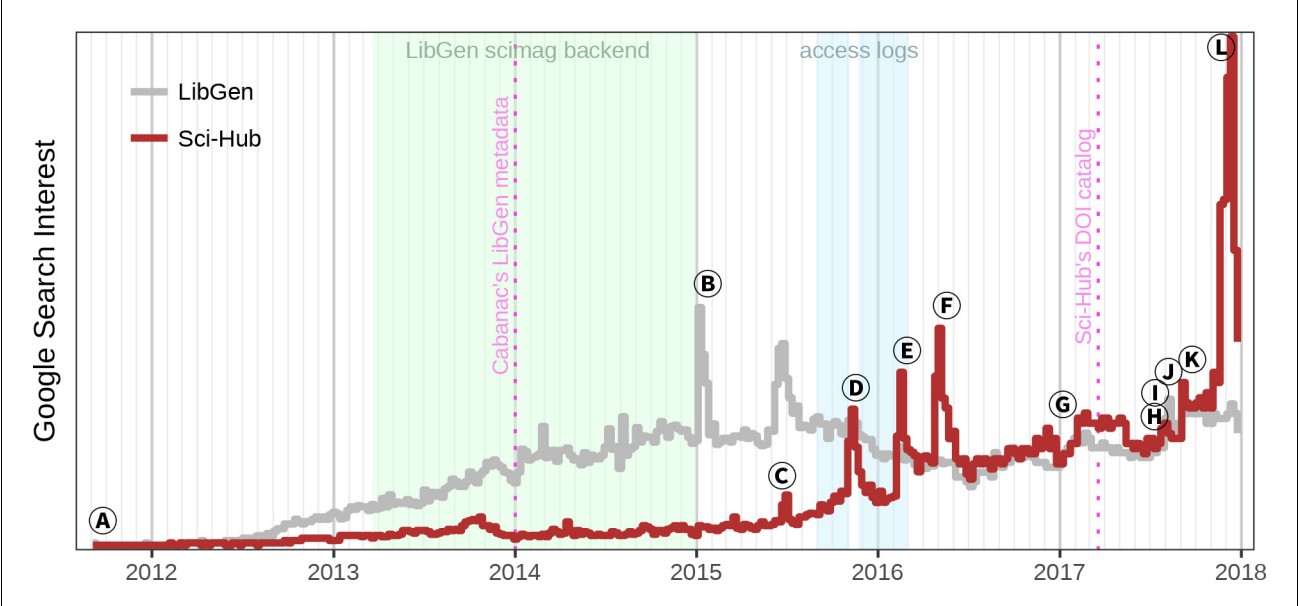

**Figure 1.** The history of Sci-Hub. Weekly interest from Google Trends is plotted over time for the search terms "Sci-Hub" and "LibGen". The light green period indicates when Sci-Hub used LibGen as its database for storing articles (***Elbakyan, 2017***). Light blue indicates the collection period of the Sci-Hub access logs that we analyze throughout this study (***Elbakyan and Bohannon, 2016***). Based on these logs and newly released logs for 2017, *Figure 1—figure supplement 1* shows the number of articles downloaded from Sci-Hub over time, providing an alternative look into Sci-Hub's growth. The first pink dotted line represents the collection date of the LibGen scimag metadata used in Cabanac's study (***Cabanac, 2016***; ***Cabanac, 2017***). The second pink dotted line shows the date of Sci-Hub's tweeted DOI catalog used in this study. The events indicated by the letters (A), (B), (C) . . . are explained in the main text.

DOI: https://doi.org/10.7554/eLife.32822.002

The following figure supplement is available for figure 1:

**Figure supplement 1.** Downloads per day on Sci-Hub for months with access logs.
DOI: https://doi.org/10.7554/eLife.32822.003

including in developed countries (***Bohannon, 2016b***). These findings spark debate among scholars, with a large contingent of scientists supporting Sci-Hub's mission (***Woolston, 2016***; ***Travis, 2016***).

○ **G** Alexandra Elbakyan is named one of "*Nature*'s 10", which featured "ten people who mattered" in 2016 (***Van Noorden, 2016***). This article profiles Alexandra and includes an estimate that Sci-Hub serves "3% of all downloads from science publishers worldwide."

○ **H** The court finds that Alexandra Elbakyan, Sci-Hub, and LibGen are "liable for willful copyright infringement" in a default judgment, since none of the defendants answered Elsevier's complaint (***Schiermeier, 2017a***; ***Van der Sar, 2017a***; ***Sweet, 2017***). The court issues a permanent injunction and orders the defendants to pay Elsevier $15 million, or $150,000 for each of 100 copyrighted works. The statutory damages, which the defendants do not intend to pay, now bear interest.

○ **I** The American Chemical Society files suit against Sci-Hub in the U.S. District Court for the Eastern District of Virginia. Their "prayer for relief" requests that Internet search engines and Internet service providers "cease facilitating access" to Sci-Hub (***Van der Sar, 2017b***; ***Barnes A et al., 2017***).

○ **J** The version 1 preprint of this study is published (***Himmelstein et al., 2017a***), generating headlines such as *Science*'s "subscription journals are doomed" (***McKenzie, 2017***) and *Inside Higher Ed*'s "Inevitably Open" (***Fister, 2017***).

○ **K** Sci-Hub blocks access to Russian IP addresses due to disputes with the Russian Scientific establishment and the naming of a newly discovered parasitoid wasp species, *Idiogramma elbakyanae*, after Alexandra Elbakyan (***Standish, 2017***; ***Khalaim and Ruíz-Cancino, 2017***). Four days later, Sci-Hub restores access after receiving "many letters of support from Russian researchers".

○ **L** The court rules on the American Chemical Society suit, ordering Sci-Hub to pay $4.8 million in damages and that "any person or entity in active concert or

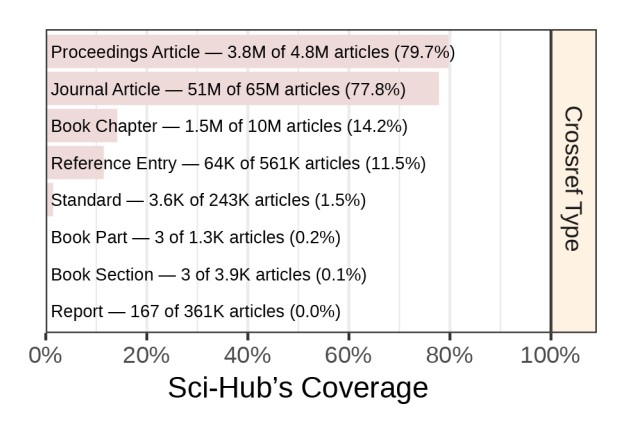

**Figure 2.** Coverage by article type. Coverage is plotted for the Crossref work types included by this study. We refer to all of these types as "articles".
DOI: https://doi.org/10.7554/eLife.32822.004

participation" with Sci-Hub "including any Internet search engines, web hosting and Internet service providers, domain name registrars, and domain name registries, cease facilitating access" (*Schiermeier, 2017b*; *Brinkema L, 2017*). Within five weeks, the domains sci-hub.io, sci-hub.ac, sci-hub.cc, and sci-hub.bz were suspended by their respective domain name registries (*Silver, 2017*), leaving only the Tor hidden service and several newly-registered/revealed domains in operation.

Past research sheds some light on Sci-Hub's reach. From the Spring of 2013 until the end of 2014, Sci-Hub relied on the Library Genesis (Lib-Gen) scimag repository to store articles (*Elbakyan, 2017*). Whenever a user requested an article, Sci-Hub would check LibGen for a copy. If the article was not in LibGen, Sci-Hub would fetch the article for the user and then upload it to LibGen. Cabanac compared the number of articles in the LibGen scimag database at the start of 2014 to the total number of Crossref DOIs, estimating that LibGen contained 36% of all published scholarly articles (*Cabanac, 2016*). Coverage was higher for several prominent publishers: 77% for Elsevier, 73% for Wiley, and 53% for Springer (prior to its merger with Macmillan/Nature; *Van Noorden, 2015*).

Later, Bohannon analyzed six months of Sci-Hub's server access logs, starting in September 2015 (*Bohannon, 2016b*). He found a global pattern of usage. Based on these logs, Gardner, McLaughlin, and Asher estimated the ratio of publisher downloads to Sci-Hub downloads within the U.S. for several publishers

(*Gardner et al., 2017*). They estimated this ratio at 20:1 for the Royal Society of Chemistry and 48:1 for Elsevier. They also noted that 25% of Sci-Hub downloads in the U.S. were for articles related to clinical medicine. Greshake also analyzed the logs to identify per capita Sci-Hub usage (*Greshake, 2016*). Portugal, Iran, Tunisia, and Greece had the highest usage, suggesting Sci-Hub is preferentially used in countries with poor institutional access to scholarly literature. In a subsequent study, he found especially high Sci-Hub usage in chemistry, with 12 of the top 20 requested journals specializing in chemistry (*Greshake, 2017a*; *Greshake, 2017b*).

Since 2015, Sci-Hub has operated its own repository, distinct from LibGen. On March 19, 2017, Sci-Hub released the list of DOIs for articles in its database. Greshake retrieved metadata for 77% of Sci-Hub DOIs (*Greshake, 2017a*; *Greshake, 2017b*). He found that 95% of articles in Sci-Hub were published after 1950. Sci-Hub requests were even more skewed towards recent articles, with only 5% targeting articles published before 1983. Greshake's study did not incorporate a catalog of all scholarly literature. This study analyzes Sci-Hub's catalog in the context of all scholarly literature and thus assesses coverage. In other words, what percentage of articles in a given domain does Sci-Hub have in its repository?

## Results

To define the extent of the scholarly literature, we relied on DOIs from the Crossref database, as downloaded on March 21, 2017. We define the "scholarly literature" as 81,609,016 texts identified by their DOIs. We refer to these texts as "articles", although Sci-Hub encompasses a range of text types, including book chapters, conference papers, and journal front matter. To assess the articles available from Sci-Hub, we relied on a list of DOIs released by Sci-Hub on March 19, 2017. All DOIs were lowercased to be congruent across datasets (see Methods). Sci-Hub's offerings included 56,246,220 articles from the corpus of scholarly literature, equating to 68.9% of all articles.

### Coverage by article type

Each article in Crossref's database is assigned a type. *Figure 2* shows coverage by article type. The scholarly literature consisted primarily of journal articles, for which Sci-Hub had 77.8% coverage. Sci-Hub's coverage was also strong for the 5 million proceedings articles at 79.7%.

**Table 1.** Coverage for the ten journals with the most articles.

| Journal | Sci-Hub | Crossref | Coverage |
|---|---|---|---|
| The Lancet | 457,650 | 458,580 | 99.8% |
| Nature | 385,619 | 399,273 | 96.6% |
| British Medical Journal (Clinical Research Edition) | 17,141 | 392,277 | 4.4% |
| Lecture Notes in Computer Science | 103,675 | 356,323 | 29.1% |
| Science | 230,649 | 251,083 | 91.9% |
| Journal of the American Medical Association | 191,950 | 248,369 | 77.3% |
| Journal of the American Chemical Society | 189,142 | 189,567 | 99.8% |
| Scientific American | 22,600 | 186,473 | 12.1% |
| New England Journal of Medicine | 180,321 | 180,467 | 99.9% |
| PLOS ONE | 4,731 | 177,260 | 2.7% |

The total number of articles published by each journal is noted in the Crossref column. The table provides the number (Sci-Hub column) and percentage (Coverage column) of these articles that are in Sci-Hub's repository.
DOI: https://doi.org/10.7554/eLife.32822.005

Overall coverage suffered from the 10 million book chapters, where coverage was poor (14.2%). The remaining Crossref types were uncommon, and hence contributed little to overall coverage.

## Coverage by journal

We defined a comprehensive set of scholarly publishing venues, referred to as "journals", based on the Scopus database. In reality, these include conferences with proceedings as well as book series. For inclusion in this analysis, each required an ISSN and at least one article as part of the Crossref-derived catalog of scholarly literature. Accordingly, our catalog consisted of 23,037 journals encompassing 56,755,671 articles. Of these journals, 4,598 (20.0%) were inactive (i.e. no longer publishing articles), and 2,933 were open access (12.7%). Only 70 journals were inactive and also open access.

We calculated Sci-Hub's coverage for each of the 23,037 journals (examples in *Table 1*). A complete journal coverage table is available in our Sci-Hub Stats Browser: https://greenelab.github.io/scihub/#/journals. The Browser also provides views for each journal and publisher with detailed coverage and access-log information.

In general, a journal's coverage was either nearly complete or near zero (*Figure 3*). As a result, relatively few journals had coverage between 5–75%. At the extremes, 2,574 journals had zero coverage in Sci-Hub, whereas 2,095 journals had perfect coverage. Of zero-coverage journals, 22.2% were inactive, and 27.9% were open access. Of perfect-coverage journals, 81.6% were inactive, and 2.0% were open

access. Hence, inactive, toll access journals make up the bulk of perfect-coverage journals.

Next, we explored article coverage according to journal attributes (*Figure 4*). Sci-Hub covered 83.1% of the 56,755,671 articles that were attributable to a journal. Articles from inactive journals had slightly lower coverage than active journals (77.3% versus 84.1%). Strikingly, coverage was substantially higher for articles from toll rather than open access journals (85.1% versus 48.3%). Coverage did vary by subject area, with the highest coverage in chemistry at 93.0% and the lowest coverage in computer science at 76.3%. Accordingly, no discipline had coverage below 75%. See *Figure 4—figure supplement 1* for coverage according to a journal's country of publication.

We also evaluated whether journal coverage varied by journal impact. We assessed journal impact using the 2015 CiteScore, which measures the average number of citations that articles published in 2012–2014 received during 2015. Highly cited journals tended to have higher coverage in Sci-Hub (Figure 9A). The 1,734 least cited journals (lowest decile) had 40.9% coverage on average, whereas the 1,733 most cited journals (top decile) averaged 90.0% coverage.

## Coverage by publisher

Next, we evaluated coverage by publisher (*Figure 5*; full table available at https://greenelab.github.io/scihub/#/publishers). The largest publisher was Elsevier, with 13,115,639 articles from 3,410 journals. Sci-Hub covered 96.9% of Elsevier articles. For the eight publishers with more than one million articles, the following coverage was observed: 96.9% of Elsevier, 89.7% of

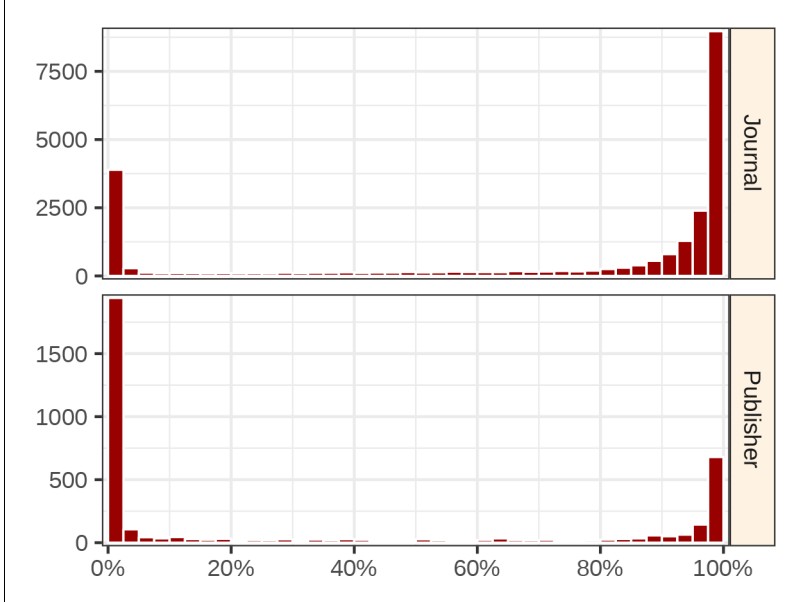

**Figure 3.** Distributions of journal & publisher coverages. The histograms show the distribution of Sci-Hub's coverage for all 23,037 journals (top) and 3,832 publishers (bottom). Each bin spans 2.5 percentage points. For example, the top-left bar indicates Sci-Hub's coverage is between 0.0%–2.5% for 3,892 journals.
DOI: https://doi.org/10.7554/eLife.32822.006

Springer Nature, 94.7% of Wiley-Blackwell, 92.6% of Taylor & Francis, 79.4% of Wolters Kluwer, 88.3% of Oxford University Press, 90.9% of SAGE, and 98.8% of American Chemical Society articles. In total, 3,832 publishers were represented in the journal catalog. The coverage distribution among publishers resembled the journal coverage distribution, with most publishers occupying the extremities (*Figure 3*). Sci-Hub had zero coverage for 1,249 publishers, and complete coverage for 341 publishers.

### Coverage by year
Next, we investigated coverage based on the year an article was published (*Figure 6*). For most years since 1850, annual coverage is between 60–80%. However, there is a dropoff in coverage, starting in 2010, for recently published articles. For example, 2016 coverage was 56.0% and 2017 coverage (for part of the year) was 45.3%. One factor is that it can take some time for Sci-Hub to retrieve articles following their publication, as many articles are not downloaded until requested by a user. Another possible factor is that some publishers are now deploying more aggressive measures to deter unauthorized article downloads (*Rovner, 2014*; *Becker, 2016*), making recent articles less accessible.

In addition, the prevalence of open access has been increasing, while Sci-Hub preferentially covers articles in toll access journals. *Figure 6—figure supplement 1* tracks yearly coverage separately for articles in toll and open access journals. Toll access coverage exceeded 80% every year since 1950 except for 2016 and 2017. For both toll and open articles, the recent dropoff in coverage appears to begin in 2014 (*Figure 6—figure supplement 1*) compared to 2010 when calculated across all articles (*Figure 6*). We speculate this discrepancy results from the proliferation of obscure, low-quality journals over the last decade (*Shen and Björk, 2015*), as these journals generally issue DOIs but are not indexed in Scopus, and therefore would be included in *Figure 6* but not in *Figure 6—figure supplement 1*. In addition to having limited readership demand, these journals are generally open access, and thus less targeted by Sci-Hub.

Sci-Hub's coverage of 2016 articles in open access journals was just 32.7% compared to 78.8% for articles in toll access journals (*Figure 6—figure supplement 1*). Upon further investigation, we discovered that in June 2015, Sci-Hub ceased archiving articles in *PeerJ*, *eLife*, and PLOS journals, although they continued archiving articles in other open access journals such as *Scientific Reports*, *Nature Communications*, and BMC-series journals. Sci-Hub currently redirects requests for these delisted journals to the publisher's site, unless it already possesses the article, in which case it serves the PDF. These findings suggest Sci-Hub prioritizes circumventing access barriers rather than creating a single repository containing every scholarly article.

### Coverage by category of access status
In the previous analyses, open access status was determined at the journal level according to Scopus. This category of access is frequently referred to as "gold" open access, meaning that all articles from the journal are available gratis. However, articles in toll access journals may also be available without charge. Adopting the terminology from the recent "State of OA" study (*Piwowar et al., 2018*), articles in toll access journals may be available gratis from the publisher under a license that permits use (termed "hybrid") or with all rights reserved (termed "bronze"). Alternatively, "green" articles are paywalled on the publisher's site, but available gratis from an open access repository (e.g. a pre- or post-print server, excluding Sci-Hub and academic social networks).

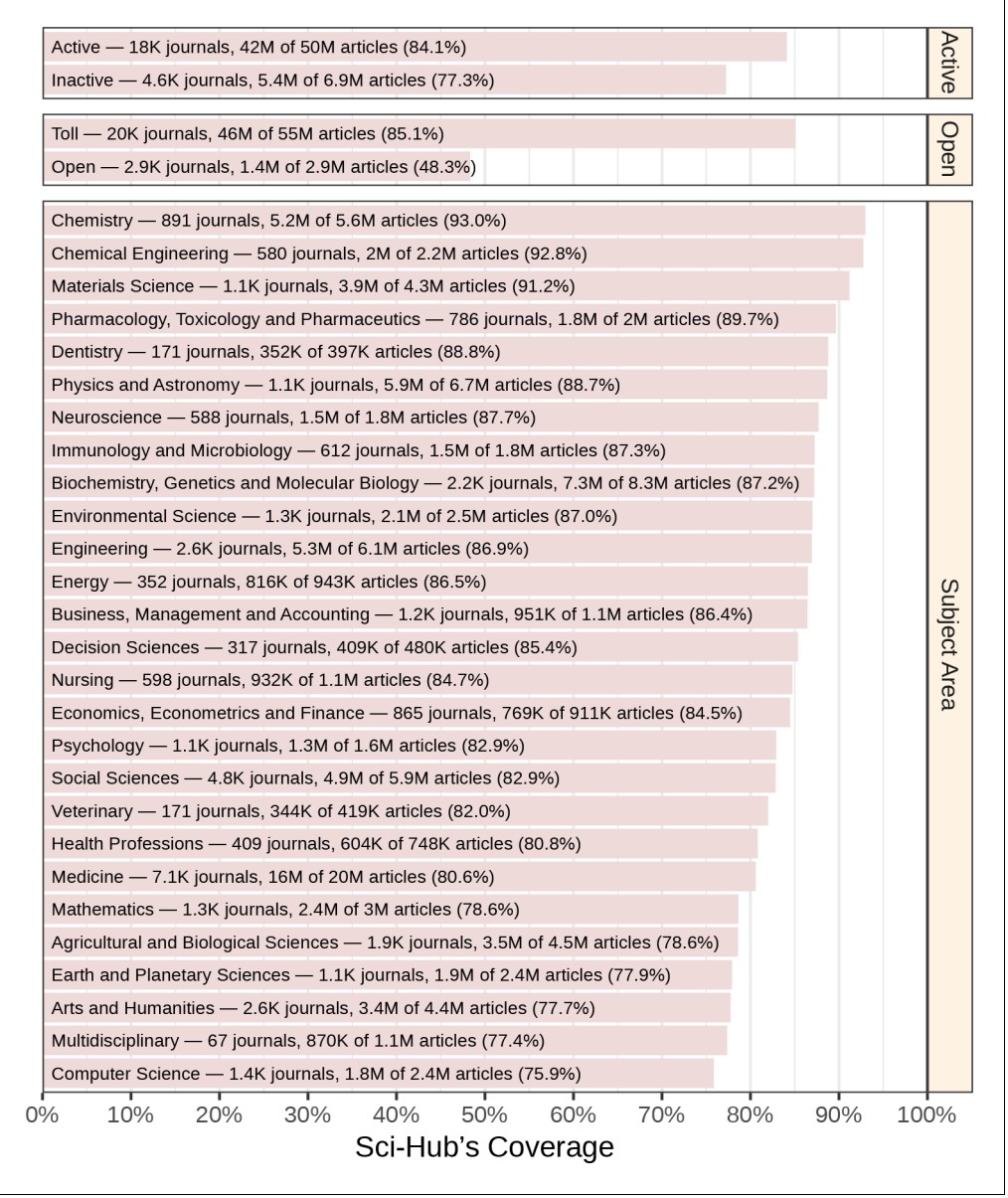

**Figure 4.** Coverage by journal attributes. Each bar represents Sci-Hub's coverage of articles in journals with the specified attributes, according to Scopus. Active refers to whether a journal still publishes articles. Open refers to whether a journal is open access. Subject area refers to a journal's discipline. Note that some journals are assigned to multiple subject areas. As an example, we identified 588 neuroscience journals, which contained 1.8 million articles. Sci-Hub possessed 87.7% of these articles.

DOI: https://doi.org/10.7554/eLife.32822.007

The following figure supplement is available for figure 4:

**Figure supplement 1.** Coverage by country of publication.

DOI: https://doi.org/10.7554/eLife.32822.008

The State of OA study determined the access status of 290,120 articles using the oaDOI utility (see Methods). *Figure 7* shows Sci-Hub's coverage for each category of access status. In line with our findings on the entire Crossref article catalog where Sci-Hub covered 49.1% of articles in open access journals, Sci-Hub's coverage of gold articles in the State of OA dataset was 49.2%. Coverage of the 165,340 closed articles was 90.4%.

Sci-Hub's coverage was higher for closed and green articles than for hybrid or bronze articles. Furthermore, Sci-Hub's coverage of closed articles was similar to its coverage of green

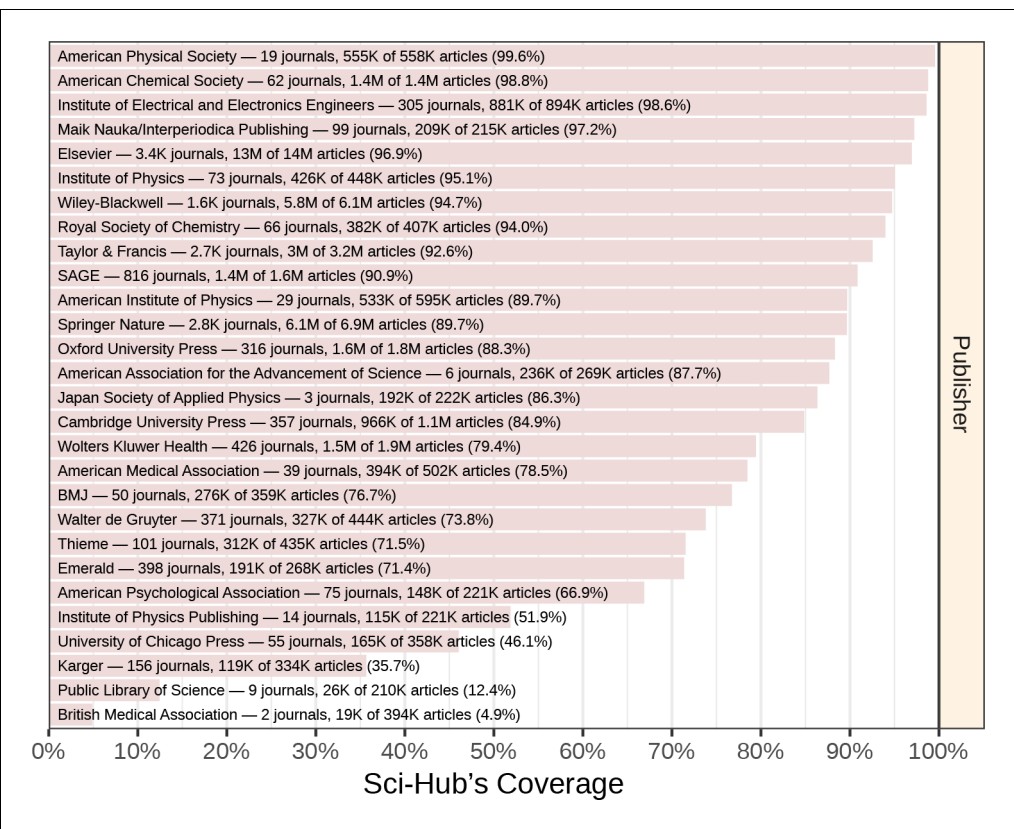

**Figure 5.** Coverage by publisher. Article coverage is shown for all Scopus publishers with at least 200,000 articles.
DOI: https://doi.org/10.7554/eLife.32822.010

articles (*Figure 7*). These findings suggest a historical pattern where users resort to Sci-Hub after encountering a paywall but before checking oaDOI or a search engine for green access. As such, Sci-Hub receives requests for green articles, triggering it to retrieve green articles at a similar rate to closed articles. However, hybrid and bronze articles, which are available gratis from their publishers, are requested and thus retrieved at a lower rate.

*Coverage of Penn Libraries*

As a benchmark, we decided to compare Sci-Hub's coverage to the access provided by a major research library. Since we were unaware of any studies that comprehensively profiled library access to scholarly articles, we collaborated with Penn Libraries to assess the extent of access available at the University of Pennsylvania (Penn). Penn is a private research university located in Philadelphia and founded by the open science pioneer Benjamin Franklin in 1749. It is one of the world's wealthiest universities, with an endowment of over $10 billion. According to the Higher Education Research and Development Survey, R&D expenditures at Penn totaled $1.29 billion in 2016, placing it third among U.S. colleges and universities. In 2017, Penn Libraries estimates that it spent $13.13 million on electronic resources, which includes subscriptions to journals and ebooks. During this year, its users accessed 7.3 million articles and 860 thousand ebook chapters, averaging a per-download cost of $1.61.

Penn Libraries uses the Alma library resource management system from Ex Libris. Alma includes an OpenURL resolver, which the Penn Libraries use to provide a service called Penn-Text for looking up scholarly articles. PennText indicates whether an article's fulltext is available online, taking into account Penn's digital subscriptions. Using API calls to PennText's OpenURL resolver, we retrieved Penn's access status for the 290,120 articles analyzed by the State of OA study (see the greenelab/library-access repository). We randomly selected 500 of these articles to evaluate manually and assessed whether their fulltexts were available from within Penn's network as well as from outside of any institutional network. We defined access as

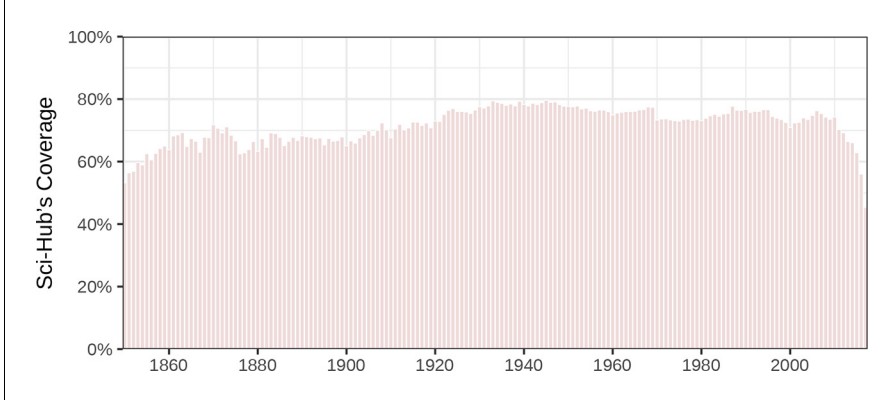

**Figure 6.** Coverage of articles by year published. Sci-Hub's article coverage is shown for each year since 1850.

DOI: https://doi.org/10.7554/eLife.32822.011

The following figure supplement is available for figure 6:

**Figure supplement 1.** Coverage of articles by year published and journal access status.

DOI: https://doi.org/10.7554/eLife.32822.012

fulltext availability at the location redirected to by an article's DOI, without providing any payment, credentials, or login information. This definition is analogous to the union of oaDOI's gold, hybrid, and bronze categories.

Using these manual access calls, we found PennText correctly classified access 88.2% [85.2%–90.8%] of the time (bracketed ranges refer to 95% confidence intervals calculated using Jeffreys interval for binomial proportions (*Rubin and Schenker, 1987*)). PennText claimed to have access to 422 of the 500 articles [81.0%–87.4%]. When PennText asserted access, it was correct 94.8% [92.4%–96.6%] of the time.

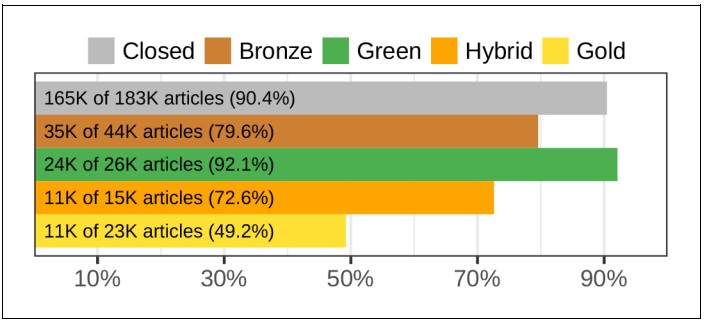

**Figure 7.** Sci-Hub's coverage by oaDOI access status. Using oaDOI calls from the State of OA study, we show Sci-Hub's coverage on each access status. Gray indicates articles that were not accessible via oaDOI (referred to as closed). Here, all three State of OA collections were combined, yielding 290,120 articles. *Figure 7—figure supplement 1* shows coverage separately for the three State of OA collections.

DOI: https://doi.org/10.7554/eLife.32822.013

The following figure supplement is available for figure 7:

**Figure supplement 1.** Coverage by oaDOI access status on each State of OA collection.

DOI: https://doi.org/10.7554/eLife.32822.014

However, when PennText claimed no access, it was only correct for 41 of 78 articles [41.6%–63.4%]. This error rate arose because PennText was not only unaware of Penn's access to 23 open articles, but also unaware of Penn's subscription access to 14 articles. Despite these issues, PennText's estimate of Penn's access at 84.4% did not differ significantly from the manually evaluated estimate of 87.4% [84.3%–90.1%]. Nonetheless, we proceed by showing comparisons for both the 500 articles with manual access calls as well as the 290,120 articles with PennText calls.

*Coverage combining access methods*

In practice, readers of the scholarly literature likely use a variety of methods for access. *Figure 8* compares several of these methods, as well as their combinations. Users without institutional access may simply attempt to view an article on its publisher's site. Based on our manual evaluation of 500 articles, we found 34.8% [30.7%–39.1%] of articles were accessible this way. The remaining 326 articles that were not accessible from their publisher's site are considered toll access. oaDOI — a utility that redirects paywalled DOIs to gratis, licit versions, when possible (*Piwowar et al., 2018*) — was able to access 15.3% [11.7%–19.5%] of these toll access articles, indicating that green open access is still limited in its reach. This remained true on the full set of 208,786 toll access articles from the State of OA dataset, where oaDOI only provided access to 12.4% [12.3%–12.6%]. Although oaDOI's overall access rate was 37.0% [36.8%–37.2%], this access consisted largely of gold, hybrid, and bronze articles, whereby gratis access is provided by the publisher.

Sci-Hub and Penn had similar coverage on all articles: 85.2% [81.9%–88.1%] versus 87.4% [84.3%–90.1%] on the manual article set and 84.8% [84.7%–84.9%] versus 84.4% [84.3%–84.5%] on the larger but automated set. However, when considering only toll access articles, Sci-Hub's coverage exceeds Penn's: 94.2% [91.2%–96.3%] versus 80.7% [76.1%–84.7%] on the manual set and 90.7% [90.5%–90.8%] versus 83.5% [83.4%–83.7%] on the automated set. This reflects Sci-Hub's focus on paywalled articles. In addition, Sci-Hub's coverage is a lower bound for its access rate, since it can retrieve articles on demand, so in practice Sci-Hub's access to toll access articles could exceed Penn's by a higher margin. Remarkably, Sci-Hub provided greater access to paywalled articles than a leading research university spending millions of

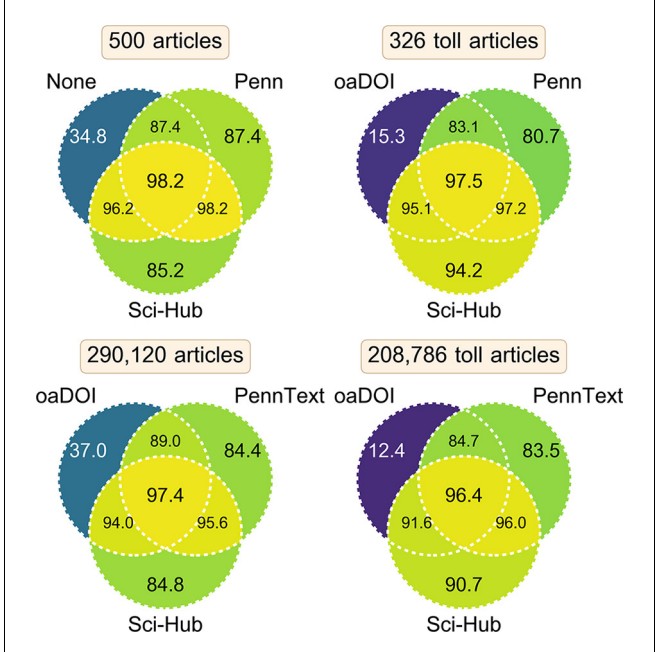

**Figure 8.** Coverage of several access methods and their combinations. This figure compares datasets of article coverage corresponding to various access methods. These article sets refer to manually evaluated access via the publisher's site from outside of an institutional network (labeled None) or from inside Penn's network (labeled Penn); access according to Penn's library system (labeled PennText); access via the oaDOI utility (labeled oaDOI); and inclusion in Sci-Hub's database (labeled Sci-Hub). Each diagram shows the coverage of three access methods and their possible combinations. Within a diagram, each section notes the percent coverage achieved by the corresponding combination of access methods. **Contrary to traditional Venn diagrams**, each section does not indicate disjoint sets of articles. Instead, each section shows coverage on the same set of articles, whose total number is reported in the diagram's title. The top two diagrams show coverage on a small set of manually evaluated articles (confidence intervals provided in the main text). The bottom two diagrams show coverage on a larger set of automatically evaluated articles. The two lefthand diagrams show coverage on all articles, whereas the two righthand diagrams show coverage on toll access articles only. Specifically, the top-right diagram assesses coverage on articles that were inaccessible from outside of an institutional network. Similarly, the bottom-right diagram assesses coverage of articles that were classified as closed or green by oaDOI, and thus excludes gold, hybrid, and bronze articles (those available gratis from their publisher).

DOI: https://doi.org/10.7554/eLife.32822.015

US dollars per year on subscriptions. However, since Sci-Hub is able to retrieve articles through many university networks, it is perhaps unsurprising that its coverage would exceed that of any single university.

Combining access methods can also be synergistic. Specifically when including open access articles, combining Sci-Hub's repository with oaDOI's or Penn's access increased coverage from around 85% to 95%. The benefits of oaDOI were reduced when only considering toll access articles, where oaDOI only improved Sci-Hub's or Penn's coverage by approximately 1%. On toll access articles, Penn's access appeared to

complement Sci-Hub's. Together, Sci-Hub's repository and Penn's access covered approximately 96% of toll access articles [95.0%–98.6% (manual set), 95.9%–96.1% (automated set)]. Our findings suggest that users with institutional subscriptions comparable to those at Penn as well as knowledge of oaDOI and Sci-Hub are able to access over 97% of all articles [96.7%–99.1% (manual set), 97.3%–97.5% (automated set)], online and without payment.

## Coverage of recently cited articles

The coverage metrics presented thus far give equal weight to each article. However, we know that article readership and by extension Sci-Hub requests are not uniformly distributed across all articles. Instead, most articles receive little readership, with a few articles receiving great readership. Therefore, we used recent citations to estimate Sci-Hub's coverage of articles weighted by user needs.

We identified 7,312,607 outgoing citations from articles published since 2015. 6,657,410 of the recent citations (91.0%) referenced an article that was in Sci-Hub. However, if only considering the 6,264,257 citations to articles in toll access journals, Sci-Hub covered 96.2% of recent citations. On the other hand, for the 866,115 citations to articles in open access journals, Sci-Hub covered only 62.3%.

## Sci-Hub access logs

Sci-Hub released article access records from its server logs, covering 165 days from September 2015 through February 2016 (*Elbakyan and Bohannon, 2016*; *Bohannon, 2016b*). After processing, the logs contained 26,984,851 access events. Hence, Sci-Hub provided access to an average of 164,000 valid requests per day in late 2015–early 2016.

In the first version of this study (*Himmelstein et al., 2017a*), we mistakenly treated the log events as requests rather than downloads. Fortunately, Sci-Hub reviewed the preprint in a series of tweets, and pointed out the error, stating "in Sci-Hub access logs released previous year, all requests are resolved requests, i.e. user successfully downloaded PDF with that DOI ... unresolved requests are not saved". Interestingly however, 198,600 access events from the logs pointed to DOIs that were not in Sci-Hub's subsequent DOI catalog. 99.1% of these events — corresponding to DOIs logged as accessed despite later being absent from Sci-Hub — were for book chapters. Upon

further investigation, we identified several DOIs in this category that Sci-Hub redirected to Lib-Gen book records as of September 2017. The LibGen landing pages were for the entire books, which contained the queried chapters, and were part of LibGen's book (not scimag) collection. The explanation that Sci-Hub outsources some book access to LibGen (and logged such requests as accessed) is corroborated by Elbakyan's statement that (*Elbakyan, 2017*): "Currently, the Sci-Hub does not store books, for books users are redirected to LibGen, but not for research papers. In future, I also want to expand the Sci-Hub repository and add books too." Nonetheless, Sci-Hub's catalog contains 72.4% of the 510,760 distinct book chapters that were accessed according to the logs. Therefore, on a chapter-by-chapter basis, Sci-Hub does already possess many of the requested scholarly books available from LibGen.

We computed journal-level metrics based on average article downloads. The "visitors" metric assesses the average number of IP addresses that accessed each article published by a journal during the 20 months preceding September 2015 (the start date of the Sci-Hub logs). In aggregate, articles from toll access journals averaged 1.30 visitors, whereas articles from open access journals averaged 0.25 visitors. *Figure 9B* shows that articles from highly cited journals were visited much more frequently on average. Articles in the least cited toll access journals averaged almost zero visitors, compared to approximately 15 visitors for the most cited journals. In addition, *Figure 9B* shows that articles in

toll access journals received many times more visitors than those in open access journals, even after accounting for journal impact. One limitation of using this analysis to judge Sci-Hub's usage patterns is that we do not know to what extent certain categories of articles were resolved (and thus logged) at different rates.

## Discussion

Sci-Hub's repository contained 69% of all scholarly articles with DOIs. Coverage for the 54.5 million articles attributed to toll access journals — which many users would not otherwise be able to access — was 85.1%. Since Sci-Hub can retrieve, in real time, requested articles that are not in its database, our coverage figures are a lower bound. Furthermore, Sci-Hub preferentially covered popular, paywalled articles. We find that 91.0% of citations since 2015 were present in Sci-Hub's repository, which increased to 96.2% when excluding citations to articles in open access journals. Journals with very low (including zero) coverage tended to be obscure, less cited venues, while average coverage of the most cited journals exceeded 90%.

We find strong evidence that Sci-Hub is primarily used to circumvent paywalls. In particular, users accessed articles from toll access journals much more frequently than open access journals. Additionally, within toll access journals, Sci-Hub provided higher coverage of articles in the closed and green categories (paywalled by the publisher) as opposed to the hybrid and bronze categories (available gratis from the publisher). Accordingly, many users likely only resort to Sci-

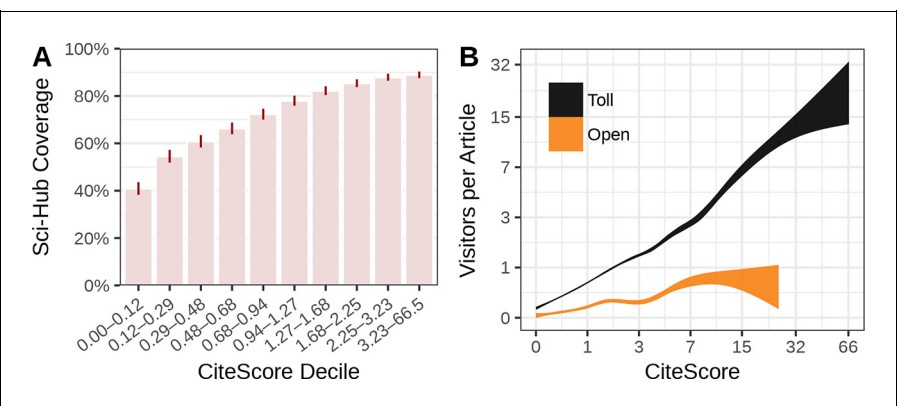

**Figure 9.** Relation to journal impact. (A) Average coverage for journals divided into 2015 CiteScore deciles. The CiteScore range defining each decile is shown by the x-axis labels. The ticks represent 99% confidence intervals of the mean. This is the only analysis where "Sci-Hub Coverage" refers to journal-level rather than article-level averages. (B) The association between 2015 CiteScore and average visitors per article is plotted for open and toll access journals. Curves show the 95% confidence band from a Generalized Additive Model.
DOI: https://doi.org/10.7554/eLife.32822.009

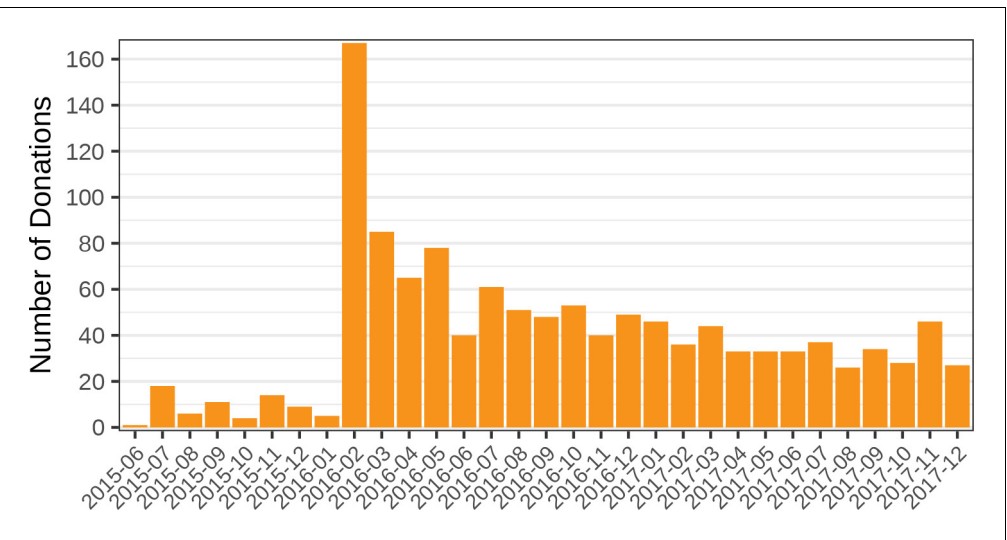

**Figure 10.** Number of bitcoin donations per month. The number of bitcoin donations to Sci-Hub is shown for each month from June 2015 to December 2017. Since February 2016, Sci-Hub has received over 25 donations per month. Each donation corresponds to an incoming transaction to a known Sci-Hub address. See *Figure 10—figure supplement 1* for the amount donated each month, valued in BTC or USD.

DOI: https://doi.org/10.7554/eLife.32822.016

The following figure supplement is available for figure 10:

**Figure supplement 1.** Bitcoin donations to Sci-Hub per month.

DOI: https://doi.org/10.7554/eLife.32822.017

---

Hub when access through a commercial database is cumbersome or costly. Finally, we observed evidence that Sci-Hub's primary operational focus is circumventing paywalls rather than compiling all literature, as archiving was deactivated in 2015 for several journals that exemplify openness. Attesting to its success in this mission, Sci-Hub's database already contains more toll access articles than are immediately accessible via the University of Pennsylvania, a leading research university.

Judging from donations, many users appear to value Sci-Hub's service. In the past, Sci-Hub accepted donations through centralized and regulated payment processors such as PayPal, Yandex, WebMoney, and QiQi (*DeMarco et al., 2015a*; *Woltermann, 2015*). Now however, Sci-Hub only advertises donation via Bitcoin, presumably to avoid banking blockades or government seizure of funds. Since the ledger of Bitcoin transactions is public, we can evaluate the donation activity to known Sci-Hub addresses (1K4t2vSBSS2xFjZ6PofYnbgZew-jeqbG1TM, 14ghuGKDAPdEcUQN4zuzGw-BUrhQgACwAyA, 1EVkHpdQ8VJQRpQ15hS-RoohCztTvDMEepm). We find that, prior to 2018, these addresses have received 1,232 donations, totaling 94.494 (*Figure 10*). Using the US dollar value at the time of transaction confirmation, Sci-Hub has received an equivalent of $69,224 in bitcoins. 85.467 bitcoins have been withdrawn from the Sci-Hub addresses via 174 transactions. Since the price of bitcoins has risen, the combined US dollar value at time of withdrawal was $421,272. At the conclusion of 2017, the Sci-Hub accounts had an outstanding balance of 9.027 bitcoins, valued at roughly $120,000. In response to this study's preprint (*Himmelstein et al., 2017a*), Sci-Hub tweeted: "the information on donations … is not very accurate, but I cannot correct it: that is confidential." Therefore, presumably, Sci-Hub has received considerable donations via alternative payment systems or to unrevealed Bitcoin addresses, which our audit did not capture. Since we do not know the identity of the depositors, another possibility would be that Sci-Hub transfered bitcoins from other addresses it controlled to the identified donation addresses.

The largest, most prominent academic publishers are thoroughly covered by Sci-Hub, and these publishers have taken note. Elsevier (whose 13.5 million works are 96.9% covered by Sci-Hub) and the American Chemical Society (whose 1.4 million works are 98.8% covered) both filed suit against Sci-Hub, despite the limited enforcement options of United States courts. The widespread gratis access that Sci-

Hub provides to previously paywalled articles calls into question the sustainability of the subscription publishing model (*McKenzie, 2017*; *Lawson, 2017b*). Avoiding biblioleaks and retaining exclusive possession of digital media may prove an insurmountable challenge for publishers (*Dunn et al., 2014*). As distributed and censorship-resistant file storage protocols mature (*Benet, 2014*; *ConsenSys, 2016*), successors to Sci-Hub may emerge that no longer rely on a centralized service. Indeed, Alexandra Elbakyan is only one individual in the larger "guerilla access" movement (*Bodó, 2016*; *Bodó, 2015*; *Laskow, 2016*), which will persist regardless of Sci-Hub's fate. As such, Sci-Hub's corpus of gratis scholarly literature may be extremely difficult to suppress.

Surveys from 2016 suggest awareness and usage of Sci-Hub was not yet commonplace (*Travis, 2016*; *Mejia et al., 2017*). However, adoption appears to be growing. According to Elbakyan, the number of Sci-Hub downloads increased from 42 million in 2015 to 75 million in 2016, equating to a 79% gain (*Van Noorden, 2016*). Comparing the search interest peaks following **D** and **L** in *Figure 1*, which both correspond to domain outages and hence existing users searching how to access Sci-Hub, we estimate annual growth of 88%. As per *Figure 1—figure supplement 1*, Sci-Hub averaged 185,243 downloads per day in January–February 2016, whereas in 2017 daily downloads averaged 458,589. Accordingly, the ratio of Sci-Hub to Penn Libraries downloads in 2017 was 20:1. In addition, adoption of Sci-Hub or similar sites could accelerate due to new technical burdens on authorized access (the flip side of anti-piracy measures) (*Davis, 2016*; *Esposito, 2016*), crackdowns on article sharing via academic social networks (*Singh Chawla, 2017a*; *Singh Chawla, 2017b*), or large-scale subscription cancellations by libraries (*Esposito, 2017*).

Historically, libraries have often canceled individual journal subscriptions or switched from bundled to à-la-carte selections (*Roth, 1990*; *Fernandez et al., 2014*; *Rogers, 2012*). More recently, library consortia have threatened wholesale cancellation of specific publishers. In 2010, Research Libraries of the UK threatened to let Elsevier contracts expire (*Bergstrom et al., 2014*; *Prosser, 2011*), while the University of California raised the possibility of boycotting Nature Publishing Group. But these disputes were ultimately resolved before major cancellations transpired. But in 2017, researchers began losing access to entire publishers. Universities in the Netherlands canceled all Oxford University Press subscriptions in May 2017 (*Else, 2017*). University of Montreal reduced its subscriptions to Taylor & Francis periodicals by 93%, axing 2,231 journals (*Gagnon, 2017*). Negotiations with Elsevier reached impasses in Germany, Peru, and Taiwan. As a result, hundreds of universities have cancelled all Elsevier subscriptions (*Schiermeier and Mega, 2016*; *Schiermeier, 2018*). These developments echo the predictions of Elsevier's attorneys in 2015 (*DeMarco et al., 2015b*): "Defendants' actions also threaten imminent irreparable harm to Elsevier because it appears that the Library Genesis Project repository may be approaching (or will eventually approach) a level of 'completeness' where it can serve as a functionally equivalent, although patently illegal, replacement for ScienceDirect."

In the worst case for toll access publishers, growing Sci-Hub usage will become both the cause and the effect of dwindling subscriptions. Librarians rely on usage metrics and user feedback to evaluate subscriptions (*Roth, 1990*). Sci-Hub could decrease the use of library subscriptions as many users find it more convenient than authorized access (*Travis, 2016*). Furthermore, librarians may receive fewer complaints after canceling subscriptions, as users become more aware of alternatives. Green open access also provides an access route outside of institutional subscription. The posting of preprints and postprints has been growing rapidly (*Piwowar et al., 2018*; *Kaiser, 2017*), with new search tools to help locate them (*Singh Chawla, 2017c*). The trend of increasing green availability is poised to continue as funders mandate postprints (*Van Noorden, 2014*) and preprints help researchers sidestep the slow pace of scholarly publishing (*Powell, 2016*). In essence, scholarly publishers may have already lost the access battle. Publishers will be forced to adapt quickly to open access publishing models. In the words of Alexandra Elbakyan (*Elbakyan, 2016b*): "The effect of long-term operation of Sci-Hub will be that publishers change their publishing models to support Open Access, because closed access will make no sense anymore."

Sci-Hub is poised to fundamentally disrupt scholarly publishing. The transition to gratis availability of scholarly articles is currently underway, and such a model may be inevitable in the long term (*Lewis, 2012*; *Sutton, 2011*; *Jha, 2012*). However, we urge the community to take this opportunity to fully liberate scholarly articles, as well as explore more constructive

business models for publishing (*Paul et al., 2017*; *Vogel, 2017*; *Logan, 2017*). Only libre access, enabled by open licensing, allows building applications on top of scholarly literature without fear of legal consequences (*Himmelstein, 2016*). For example, fulltext mining of scholarly literature is an area of great potential (*Westergaard et al., 2017*), but is currently impractical due to the lack of a large-scale pre-processed corpus of articles. The barriers here are legal, not technological (*Brook et al., 2014*; *Van Noorden, 2012*). In closing, were all articles libre, there would be no such thing as a "pirate website" for accessing scholarly literature.

## Methods

This project was performed entirely in the open, via the GitHub repository greenelab/scihub. Several authors of this study became involved after we mentioned their usernames in GitHub discussions. This project's fully transparent and online model enabled us to assemble an international team of individuals with complementary expertise and knowledge.

We managed our computational environment using Conda, allowing us to specify and install dependencies for both Python and R. We performed our analyses using a series of Jupyter notebooks. In general, data integration and manipulation were performed in Python 3, relying heavily on Pandas, while plotting was performed with ggplot2 in R. Tabular data were saved in TSV (tab-separated values) format, and large datasets were compressed using XZ. We used Git Large File Storage (Git LFS) to track large files, enabling us to make nearly all of the datasets generated and consumed by the analyses available to the public. The Sci-Hub Stats Browser is a single-page application built using React and hosted via GitHub Pages. Frontend visualizations use Vega-Lite (*Satyanarayan et al., 2017*). Certain datasets for the browser are hosted in the greenelab/scihub-browser-data repository.

The manuscript source for this study is located at greenelab/scihub-manuscript. We used the Manubot to automatically generate the manuscript from Markdown files. This system — originally developed for the Deep Review to enable collaborative writing on GitHub (*Ching et al., 2017*) — uses continuous analysis to fetch reference metadata and rebuild the manuscript upon changes (*Beaulieu-Jones and Greene, 2017*).

### Digital object identifiers

We used DOIs (Digital Object Identifiers) to uniquely identify articles. The Sci-Hub and Lib-Gen scimag repositories also uniquely identify articles by their DOIs, making DOIs the natural primary identifier for our analyses. The DOI initiative began in 1997, and the first DOIs were registered in 2000 (*International DOI Foundation, 2017*; *Wang, 2007*). Note that DOIs can be registered retroactively. For example, Antony van Leewenhoeck's discovery of protists and bacteria — published in 1677 by *Philosophical Transactions of the Royal Society of London* (*van Leewenhoeck, 1677*) — has a DOI (10.1098/rstl.1677.0003), retroactively assigned in 2006.

Not all scholarly articles have DOIs. By evaluating the presence of DOIs in other databases of scholarly literature (such as PubMed, Web of Science, and Scopus), researchers estimate around 90% of newly published articles in the sciences have DOIs (*Gorraiz et al., 2016*; *Boudry and Chartron, 2017*). The prevalence of DOIs varies by discipline and country of publication, with DOI assignment in newly published Arts & Humanities articles around 60% (*Gorraiz et al., 2016*). Indeed, DOI registration is almost entirely lacking for publishers from many Eastern European countries (*Boudry and Chartron, 2017*). In addition, the prevalence of DOI assignment is likely lower for older articles (*Boudry and Chartron, 2017*). The incomplete and non-random assignment of DOIs to scholarly articles is a limitation of this study. However, DOIs are presumably the least imperfect and most widespread identifier for scholarly articles.

An often overlooked aspect of the DOI system is that DOIs are case-insensitive within the ASCII character range (*International DOI Foundation, 2017*; *British Standards Institute, 2012*). In other words, 10.7717/peerj.705 refers to the same article as 10.7717/PeerJ.705. Accordingly, DOIs make a poor standard identifier unless they are consistently cased. While the DOI handbook states that "all DOI names are converted to upper case upon registration" (*International DOI Foundation, 2017*), we lowercased DOIs in accordance with Crossref's behavior. Given the risk of unmatched DOIs, we lowercased DOIs for each input resource at the earliest opportunity in our processing pipeline. Consistent casing considerably influenced our findings as different resources used different casings of the same DOI.

### Crossref-derived catalog of scholarly articles

To catalog all scholarly articles, we relied on the Crossref database. Crossref is a DOI Registration Agency (an entity capable of assigning DOIs) for scholarly publishing (*Lammey, 2014*). There are presently 10 Registration Agencies. We estimate that Crossref has registered 67% of all DOIs in existence. While several Registration Agencies assign DOIs to scholarly publications, Crossref is the preeminent registrar. In March 2015, of the 1,464,818 valid DOI links on the English version of Wikipedia, 99.9% were registered with Crossref (*Kikkawa et al., 2016*). This percentage was slightly lower for other languages: 99.8% on Chinese Wikipedia and 98.0% on Japanese Wikipedia. Hence, the overwhelming majority of DOI-referenced scholarly articles are registered with Crossref. Since Crossref has the most comprehensive and featureful programmatic access, there was a strong incentive to focus solely on Crossref-registered DOIs. Given Crossref's preeminence, the omission of other Registration Agencies is unlikely to substantially influence our findings.

We queried the works endpoint of the Crossref API to retrieve the metadata for all DOIs, storing the responses in a MongoDB database. The queries began on March 21, 2017 and took 12 days to complete. In total, we retrieved metadata for 87,542,370 DOIs, corresponding to all Crossref works as of March 21, 2017. The source code for this step is available on GitHub at greenelab/crossref. Due to its large file size (7.4 GB), the MongoDB database export of DOI metadata is not available on GitHub, and is instead hosted via figshare (*Himmelstein et al., 2017b*). We created TSV files with the minimal information needed for this study: First, a DOI table with columns for work type and date issued. Date issued refers to the earliest known publication date, i.e. the date of print or online publication, whichever occurred first. Second, a mapping of DOI to ISSN for associating articles with their journal of publication.

We selected a subset of Crossref work types to include in our Sci-Hub coverage analyses that corresponded to scholarly articles (i.e. publications). Since we could not locate definitions for the Crossref types, we used our best judgment and evaluated sample works of a given type in the case of uncertainty. We included the following types: book-chapter, book-part, book-section, journal-article, proceedings-article, reference-entry, report, and standard. Types such as book, journal, journal-issue, and report-series were excluded, as they are generally containers for individual articles rather than scholarly articles themselves. After filtering by type, 81,609,016 DOIs remained (77,201,782 of which had their year of publication available). For the purposes of this study, these DOIs represent the entirety of the scholarly literature.

### Scopus-derived catalog of journals

Prior to June 2017, the Crossref API had an issue that prevented exhaustively downloading journal metadata. Therefore, we instead relied on the Scopus database to catalog scholarly journals. Scopus uses "title" to refer to all of the following: peer-reviewed journals, trade journals, book series, and conference proceedings. For this study, we refer to all of these types as journals. From the October 2017 data release of Scopus titles, we extracted metadata for 72,502 titles including their names, ISSNs, subject areas, publishers, open access status, and active status. The publisher information was poorly standardized — e.g. both "ICE Publishing" and "ICE Publishing Ltd." were present — so name variants were combined using OpenRefine. This version of Scopus determined open access status by whether a journal was registered in DOAJ or ROAD as of April 2017. Note that Scopus does not index every scholarly journal (*Mongeon and Paul-Hus, 2015*), which is one reason why 30.5% of articles (24,853,345 DOIs) were not attributable to a journal.

We tidied the Scopus Journal Metrics, which evaluate journals based on the number of citations their articles receive. Specifically, we extracted a 2015 CiteScore for 22,256 titles, 17,336 of which were included in our journal catalog. Finally, we queried the Elsevier API to retrieve homepage URLs for 20,992 Scopus titles. See dhimmel/scopus for the source code and data relating to Scopus.

### LibGen scimag's catalog of articles

Library Genesis (LibGen) is a shadow library primarily comprising illicit copies of academic books and articles. Compared to Sci-Hub, the operations of LibGen are more opaque, as the contributors maintain a low profile and do not contact journalists (*Elbakyan, 2017*). LibGen hosts several collections, including distinct repositories for scientific books and textbooks, fiction books, and comics (*Cabanac, 2016*). In 2012, LibGen added the "scimag" database for scholarly literature. Since the spring of 2013, Sci-

Hub has uploaded articles that it obtains to Lib-Gen scimag (*Elbakyan, 2017*). At the end of 2014, Sci-Hub forked LibGen scimag and began managing its own distinct article repository.

We downloaded the LibGen scimag meta-data database on April 7, 2017 as a SQL dump. We imported the SQL dump into MySQL, and then exported the scimag table to a TSV file (*Himmelstein and McLaughlin, 2017*). Each row of this table corresponds to an article in LibGen, identified by its DOI. The TimeAdded field apparently indicates when the publication was uploaded to LibGen. After removing records missing TimeAdded, 64,195,940 DOIs remained. 56,205,763 (87.6%) of the DOIs were in our Crossref-derived catalog of scholarly literature. The 12.4% of LibGen scimag DOIs missing from our Crossref catalog likely comprise incorrect DOIs, DOIs whose metadata availability post-dates our Crossref export, DOIs from other Registration Agencies, and DOIs for excluded publication types.

Next, we explored the cumulative size of Lib-Gen scimag over time according to the Time-Added field (*Figure 11*). However, when we compared our plot to one generated from the LibGen scimag database SQL dump on January 1, 2014 (*Cabanac, 2016*; *Cabanac, 2017*), we noticed a major discrepancy. The earlier analysis identified a total of 22,829,088 DOIs, whereas we found only 233,707 DOIs as of January 1, 2014. We hypothesize that the discrepancy arose because TimeAdded indicates the date modified rather than created. Specifically, when an article in the database is changed, the data-base record for that DOI is entirely replaced. Hence, the TimeAdded value is effectively over-written upon every update to a record. Unfortu-nately, many research questions require the date first added. For example, lag-time analyses (the time from study publication to LibGen upload) may be unreliable. Therefore, we do not report on these findings in this manuscript. Instead, we provide *Figure 11—figure supplement 1* as an example analysis that would be highly informa-tive were reliable creation dates available. In addition, findings from some previous studies may require additional scrutiny. For example, Cabanac writes (*Cabanac, 2016*): "The growth of LibGen suggests that it has benefited from a few isolated, but massive, additions of scientific articles to its cache. For instance, 71% of the article collection was uploaded in 13 days at a rate of 100,000+ articles a day. It is likely that such massive collections of articles result from biblioleaks (*Dunn et al., 2014*), but one can only

speculate about this because of the undocu-mented source of each file cached at LibGen." While we agree this is most likely the case, con-firmation is needed that the bulk addition of articles does not simply correspond to bulk updates rather than bulk initial uploads.

### Sci-Hub's catalog of articles

On March 19, 2017, Sci-Hub tweeted: "If you like the list of all DOI collected on Sci-Hub, here it is: sci-hub.cc/downloads/doi.7z . . . 62,835,101 DOI in alphabetical order". The tweet included a download link for a file with the 62,835,101 DOIs that Sci-Hub claims to provide access to. Of these DOIs, 56,246,220 were part of the Crossref-derived catalog of scholarly articles, and 99.5% of the DOIs from Sci-Hub's list were in the LibGen scimag repository (after filtering). Hence, the LibGen scimag and Sci-Hub reposito-ries have largely stayed in sync since their split. On Twitter, the Sci-Hub account confirmed this finding, commenting "with a small differences, yes the database is the same". Therefore, the LibGen scimag and Sci-Hub DOI catalogs can essentially be used interchangeably for research purposes.

### State of OA datasets

oaDOI, short for open access DOI, is a service that determines whether a DOI is available gratis somewhere online (*Piwowar, 2016*). oaDOI does not index articles posted to academic social networks or available from illicit reposito-ries such as Sci-Hub (*Piwowar et al., 2018*). Using the oaDOI infrastructure, the State of OA study investigated the availability of articles from three collections (*Piwowar et al., 2018*). Each collection consists of a random sample of approximately 100,000 articles from a larger cor-pus. We describe the collections below and report the number of articles after intersection with our DOI catalog:

- **Web of Science**: 103,491 articles pub-lished between 2009–2015 and classified as citable items in Web of Science.
- **Unpaywall**: 87,322 articles visited by Unpaywall users from June 5–11, 2017.
- **Crossref**: 99,952 articles with Crossref type of journal-article.

Unpaywall is a web-browser extension that notifies its user if an article is available via oaDOI (*Singh Chawla, 2017d*). Since the Unpaywall col-lection is based on articles that users visited, it's a better reflection of the actual access needs of contemporary scholars. Unfortunately, since the

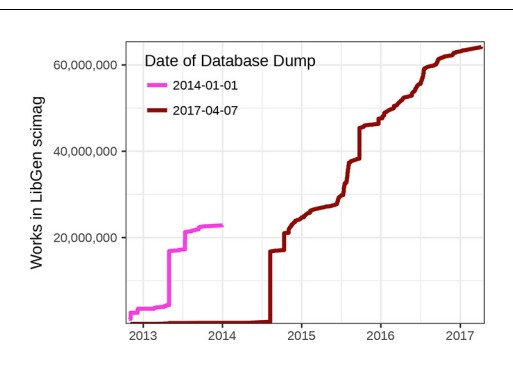

**Figure 11.** Number of articles in LibGen scimag over time. The figure shows the number of articles in LibGen scimag, according to its TimeAdded field, for two database dumps. The number of articles added per day for the January 1, 2014 LibGen database dump was provided by Cabanac and corresponds to *Figure 1* of (*Cabanac, 2016*). Notice the major discrepancy whereby articles from the April 7, 2017 database dump were added at later dates. Accordingly, we hypothesize that the TimeAdded field is replaced upon modification, making it impossible to assess date of first upload.

DOI: https://doi.org/10.7554/eLife.32822.018

The following figure supplement is available for figure 11:

**Figure supplement 1.** Lag-time from publication to LibGen upload.

DOI: https://doi.org/10.7554/eLife.32822.019

number of visits per article is not preserved by this dataset, fulfillment rate estimates are biased against highly-visited articles and become scale-variant (affected by the popularity of Unpaywall).

The State of OA study ascertained the accessibility status of each DOI in each collection using oaDOI (*Piwowar et al., 2018*; *Piwowar et al., 2017*). Articles for which oaDOI did not identify a fulltext were considered "closed". Otherwise, articles were assigned a color/status of bronze, green, hybrid, or gold. oaDOI classifies articles not available from their publisher's site as either green or closed. The version of oaDOI used in the State of OA study identified green articles by searching PubMed Central and BASE. Readers should note that this implementation likely undercounts green articles, especially if considering articles available from academic social networks as green.

### Recent citation catalog

OpenCitations is an public domain resource containing scholarly citation data (*Peroni et al., 2015*). OpenCitations extracts its information

from the Open Access Subset of PubMed Central. In the greenelab/opencitations repository, we processed the July 25, 2017 OpenCitations data release (*Peroni and Shotton, 2016*; *OpenCitations, 2017a*), creating a DOI–cites–DOI catalog of bibliographic references. For quality control, we removed DOIs that were not part of the Crossref-derived catalog of articles. Furthermore, we removed outgoing citations from articles published before 2015. Incoming citations to articles predating 2015 were not removed. The resulting catalog consisted of 7,312,607 citations from 200,206 recent articles to 3,857,822 referenced articles.

### Sci-Hub access logs

The 2016 study titled "Who's downloading pirated papers? Everyone" analyzed a dataset of Sci-Hub access logs (*Bohannon, 2016b*). Alexandra Elbakyan worked with journalist John Bohannon to produce a dataset of Sci-Hub's resolved requests from September 1, 2015 through February 29, 2016 (*Elbakyan and Bohannon, 2016*). In November 2015, Sci-Hub's domain name was suspended as the result of legal action by Elsevier (*Schiermeier, 2015*; *Van der Sar, 2015c*). According to Bohannon, this resulted in "an 18-day gap in the data starting November 4, 2015 when the domain sci-hub.org went down and the server logs were improperly configured." We show this downtime in *Figure 1*.

We filtered the access events by excluding DOIs not included in our literature catalog and omitting records that occurred before an article's publication date. This filter preserved 26,984,851 access events for 10,293,836 distinct DOIs (97.5% of the 10,552,418 distinct prefiltered DOIs). We summarized the access events for each article using the following metrics:

1. downloads: total number of times the article was accessed
2. visitors: number of IP addresses that accessed the article
3. countries: number of countries (geolocation by IP address) from which the article was accessed
4. days: number of days on which the article was accessed
5. months: number of months in which the article was accessed

Next, we calculated journal-level access metrics based on articles published from January 1, 2014 until the start of the Sci-Hub access log records on September 1, 2015. For each journal, we calculated the average values for the five

access log metrics described above. Interestingly, the journal *Medicine - Programa de Formación Médica Continuada Acreditado* received the most visitors per article, averaging 33.4 visitors for each of its 326 articles.

Note that these analyses do not include Sci-Hub's access logs for 2017 (*Tzovaras, 2018*), which were released on January 18, 2018. Unfortunately, at that time we had already adopted a freeze on major new analyses. Nonetheless, we did a quick analysis to assess growth in Sci-Hub downloads over time that combined the 2015–2016 and 2017 access log data (*Figure 1—figure supplement 1*).

## Data Availability

The source code data analysis and interactive browser associated with this study are available at the following GitHub repositories: https://github.com/greenelab/crossref (copy archived at https://github.com/elifesciences-publications/crossref)
https://github.com/greenelab/scihub (copy archived at https://github.com/elifesciences-publications/scihub)
https://github.com/greenelab/scihub-manuscript (copy archived at https://github.com/elifesciences-publications/scihub-manuscript)
https://github.com/greenelab/scihub-browser-data (copy archived at https://github.com/elifesciences-publications/sciub-brower-data)
https://github.com/dhimmel/scopus (copy archived at https://github.com/elifesciences-publications/scopus)
https://github.com/greenelab/library-access (copy archived at https://github.com/elifesciences-publications/library-access)

The MongoDB database export of DOI metadata from the Crossref API are available on Figshare (https://doi.org/10.6084/m9.figshare.4816720.v1).

### Acknowledgements

We'd like to thank the individuals, not listed as authors, who provided comments on GitHub issues or pull requests. Specifically, Ross Mounce, Richard Smith-Unna, Guillaume Cabanac, and Stuart Taylor provided valuable input while the study was underway. In addition, we're grateful to GitHub for offering gratis Large File Storage as part of their education program.

**Daniel S Himmelstein** is in the Department of Systems Pharmacology and Translational Therapeutics, University of Pennsylvania, Philadelphia, United States

daniel.himmelstein@gmail.com
https://orcid.org/0000-0002-3012-7446

**Ariel Rodriguez Romero** is at Bidwise, Inc, Miami, United States
https://orcid.org/0000-0003-2290-4927

**Jacob G Levernier** is in Library Technology Services and Strategic Initiatives, University of Pennsylvania, Philadelphia, United States
https://orcid.org/0000-0003-1563-7314

**Thomas Anthony Munro** is in the School of Life and Environmental Sciences, Deakin University, Melbourne, Australia
https://orcid.org/0000-0002-3366-7149

**Stephen Reid McLaughlin** is in the School of Information, University of Texas at Austin, Austin, United States
https://orcid.org/0000-0002-9888-3168

**Bastian Greshake Tzovaras** is in the Department of Applied Bioinformatics, Institute of Cell Biology and Neuroscience, Goethe University, Frankfurt, Germany
https://orcid.org/0000-0002-9925-9623

**Casey S Greene** is in the Department of Systems Pharmacology & Translational Therapeutics, University of Pennsylvania, Philadelphia, United States
greenescientist@gmail.com
https://orcid.org/0000-0001-8713-9213

*Author contributions:* Daniel S Himmelstein, Conceptualization, Data curation, Software, Formal analysis, Supervision, Validation, Investigation, Visualization, Methodology, Writing—original draft, Project administration, Writing—review and editing; Ariel Rodriguez Romero, Software, Investigation, Visualization, Methodology; Jacob G Levernier, Data curation, Software, Formal analysis, Validation, Investigation, Methodology, Writing—review and editing; Thomas Anthony Munro, Data curation, Methodology, Writing—review and editing; Stephen Reid McLaughlin, Conceptualization, Methodology, Writing—review and editing; Bastian Greshake Tzovaras, Data curation, Formal analysis, Investigation, Methodology, Writing—review and editing; Casey S Greene, Conceptualization, Resources, Supervision, Funding acquisition, Methodology, Project administration, Writing—review and editing

*Competing interests:* The authors declare that no competing interests exist.

**Decision letter and Author response**
Decision letter https://doi.org/10.7554/eLife.32822.021
Author response https://doi.org/10.7554/eLife.32822.022

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
