## [Decision Letter]

Thank you for submitting your article "Sci-Hub provides access to nearly all scholarly literature" to *eLife* for consideration as a Feature Article. Your article has been reviewed by three peer reviewers and the evaluation has been overseen by Peter Rogers as the *eLife* Features Editor.

The following individual involved in review of your submission has agreed to reveal their identity: David Prosser (peer reviewer).

Overall the reviewers were positive about the article, but they had a small number of concerns. We therefore invite you to prepare a revised submission that addresses these concerns.

Summary:

This is an interesting and original piece of work that focusses on one of the most potentially disruptive forces in scholarly communication in the 21st Century. It outlines, for I think the first time, the shear size and scope of Sci-Hub and the amount of material that is now available through it. It is perhaps a little early to say with any certainly what the effect of Sci-Hub will be on subscriptions and I am pleased that the authors have highlighted the issue, but not moved too far into the space of prediction. They have allowed the analysis to speak for itself.

Essential revisions:

1) The authors make one claim that seems to me not supported by the evidence. They claim that their paper shows that toll-based publishing is becoming unsustainable. But they also point to a recent study that estimates that the ratio of the number of times papers are downloaded from the publisher to downloaded from Sci-Hub is 48:1 for Elsevier and 20:1 for Royal Society of Chemistry. This suggests that Sci-Hub so far has very little influence on the subscription demand for journal articles.

2) To calculate coverage, DOIs from Crossref and Sci-Hub for over 56M records are compared. Please give details of the algorithm used to compare the DOIs.

A straight comparison of 56M DOI's with another similarly large number of DOIs would require at least 2.5 X 10^14^ comparisons. Was it done this way? If so, how long did it take? Was another clustering method used?

And if such comparisons were made, why are the percentages of coverage only given to third order? Is this just for convenience? (Giving the percentages of coverage to third order seems to imply a sample and not a complete comparison).

3) I believe if the authors compare Figure 4 with Table 2 in Piwowar et al., 2017, a strong negative correlation can be observed whereby the more likely a paper is to be found on the web, the less likely it is in Sci-Hub.

---

## [Author Response]

We'd like to thank the reviewers and editors for their feedback, which we have aimed to address in our revised manuscript, corresponding to version 3 of the study's preprint. Sci-Hub and scholarly publishing are both rapidly developing topics. As such, the revised manuscript contains additions resulting from new developments since version 2. These include the judgment in the ACS suit and the subsequent suspension of four Sci-Hub domains; the release of Sci-Hub download logs for 2017, which we visualize in Figure 1—figure supplement 1; and additional withdrawals from Sci-Hub's bitcoin addresses. As such, we extended the Google trends and Bitcoin analyses through the end of 2017. The revised version contains many enhancements, of which the major ones are summarized here.

We updated our journal information to the October 2017 release of Scopus. In addition, we created patches to standardize publisher names in Scopus. As a result, many of the numbers reported in the manuscript have changed slightly. Thomas Munro assisted with the Scopus enhancements, as well as providing feedback in other areas, and has been added as an author.

We investigated the level of access provided by the University of Pennsylvania Libraries. This is a major addition to the study, since it allows us to compare Sci-Hub's database with the access of a major research university. Furthermore, the data for this analysis are openly available, making it the most comprehensive public analysis of a university's coverage of the scholarly literature (that we're aware of). Jacob Levernier, who helped perform these analyses, has been added as an author.

[…] *Essential revisions:*

1) The authors make one claim that seems to me not supported by the evidence. They claim that their paper shows that toll-based publishing is becoming unsustainable. But they also point to a recent study that estimates that the ratio of the number of times papers are downloaded from the publisher to downloaded from Sci-Hub is 48:1 for Elsevier and 20:1 for Royal Society of Chemistry. This suggests that Sci-Hub so far has very little influence on the subscription demand for journal articles.

In light of this reviewer feedback, we recently discussed the effect of Sci-Hub on the toll access business model in greater depth. While there is disagreement on the matter, we have added additional evidence and reasoning to the manuscript regarding these claims.

The 48:1 and 20:1 estimates from Gardner et al. assess Sci-Hub's usage at the start of 2016. These numbers are, of course, small. However, awareness of Sci-Hub was also low at that time. For example, a survey conducted in the first half of 2016, found that only 19% of Latin American medical students were aware of Sci-Hub. An online survey by *Science* in May 2016 found 59% of 10,874 respondents had used Sci-Hub, although it noted "the survey sample is likely biased heavily toward fans of the site". Incidentally, the 62% of respondents thought "Sci-Hub will disrupt the traditional science publishing industry".

An important factor here is the rate at which Sci-Hub adoption will grow. Our manuscript now includes two distinct metrics for assessing Sci-Hub's annual growth: 79% according to download statistics provided by Sci-Hub and 88% according to peak Google search interest following service outage. In addition, Sci-Hub released access logs for 2017, which we analyze in Figure 1—figure supplement 1 to show how Sci-Hub downloads per days have increased over time. We observe rapid growth in 2017, peaking in July 2017 with an average of 593,880 downloads per day. Combining the new Sci-Hub log data with information from the Penn Libraries, we calculate that Sci-Hub usage for electronic access to scholarly articles exceeds Penn's usage by a ratio of 20:1. We now discuss how Sci-Hub usage could lead to subscription cancellations by affecting the usage metrics and feedback librarians use to evaluate subscriptions. Finally, we discuss the growth of green open access, composed of preprints and postprints, which may also diminish the need for subscription access.

These developments are happening in the context of library budgets that are increasingly burdened by subscription costs, as mentioned in our Introduction. Since our manuscript submission, SPARC released a Big Deal Cancellation Tracker, which supports our observation that large-scale subscription cancellations are becoming more prevalent. We now reference this resource as well as the latest coverage of the Project DEAL negations with Elsevier in Germany. We also cover the domain name suspensions following the ACS suit judgment and Sci-Hub's re-emergence at several new domains. We modified a few sentences to make it clear that our assessment of Sci-Hub's disruptive influence is a prediction and not a certainty.

2) To calculate coverage, DOIs from Crossref and Sci-Hub for over 56M records are compared. Please give details of the algorithm used to compare the DOIs.A straight comparison of 56M DOI's with another similarly large number of DOIs would require at least 2.5 X 10^14^ comparisons. Was it done this way? If so, how long did it take? Was another clustering method used?And if such comparisons were made, why are the percentages of coverage only given to third order? Is this just for convenience? (Giving the percentages of coverage to third order seems to imply a sample and not a complete comparison).

We thank the reviewers for their interest in the computational methods used for our coverage calculations. Python includes a set data type to store unordered collections with no duplicate elements. Set operations enable efficient membership testing and intersection with other sets. As a result, we do not need to apply a list intersection algorithm, as described by the reviewers, that would scale quadratically due to nested iteration.

We evaluate each Crossref DOI for membership in the set of Sci-Hub DOIs in 01.catalog-dois.ipynb. The specific implementation we apply passes a set of Sci-Hub DOIs to the pandas.Series.isin function. This function uses a hash table, the formative data structure behind sets, to enable an efficient comparison of DOIs. The 01.catalog-dois.ipynb notebook does take considerable time (perhaps up to an hour) to execute. However, much of this time is spend on file input/output, which requires reading/writing to disk and (de)compressing streams, both of which can be runtime intensive.

We create a tabular file doi.tsv.xz that contains a binary indicator (coded as 0/1) of whether each Crossref DOI is in Sci-Hub's repository. The general workflow we followed to compute coverage was to subset the DOI dataset for the relevant DOIs (e.g. all DOIs published in 2016) and take the sum/mean of the binary membership column. These operations made heavy use of pandas.DataFrame.join to incorporate additional information about DOIs and pandas.DataFrame.groupby to compute coverage for all possible subsets at once.

3) I believe if the authors compare Figure 4 with Table 2 in Piwowar et al., 2017, a strong negative correlation can be observed whereby the more likely a paper is to be found on the web, the less likely it is in Sci-Hub.

We believe this comment is comparing Figure 4 of the OA study (titled "Percentage of different access types […] per NSF discipline", also see Figure A5) with Figure 4 of our study (titled "Coverage by journal attributes"). For example, among plotted disciplines, Sci-Hub coverage is highest for chemistry, whereas oaDOI's coverage is lowest for chemistry. On the other hand, oaDOI exceeds 50% coverage of mathematics, whereas mathematics coverage is relatively low in Sci-Hub.

Regarding the hypothesis, "the more likely a paper is to be found on the web, the less likely it is in Sci-Hub", we evaluate this directly in the "Coverage by category of access status" section. This hypothesis appears true for most types of gratis web access. Compared to closed articles, bronze, hybrid, and gold OA articles are less likely to be in Sci-Hub. However, green articles (available without charge, but not from the publisher) do not seem have lower coverage than closed articles in Sci-Hub. The conclusion from the reviewers' comparison could therefore be rephrased as "the less gratis availability of a discipline's articles on the web (according to oaDOI), the greater the coverage on Sci-Hub." One possibility is that researchers from disciplines with poor oaDOI coverage more frequently encounter access problems, leading to greater awareness and usage of Sci-Hub.